# The Impact of Individualized Hemodynamic Management on Intraoperative Fluid Balance and Hemodynamic Interventions during Spine Surgery in the Prone Position: A Prospective Randomized Trial

**DOI:** 10.3390/medicina58111683

**Published:** 2022-11-20

**Authors:** Lucie Kukralova, Vlasta Dostalova, Miroslav Cihlo, Jaroslav Kraus, Pavel Dostal

**Affiliations:** 1Department of Anaesthesiology and Intensive Care Medicine, Charles University, Faculty of Medicine in Hradec Kralove, University Hospital Hradec Kralove, Sokolska 581, 500 05 Hradec Kralove, Czech Republic; 2Department of Neurosurgery, Charles University, Faculty of Medicine in Hradec Kralove, University Hospital Hradec Kralove, Sokolska 581, 500 05 Hradec Kralove, Czech Republic; 3Department of Ortopedic Surgery, J.E. Purkinje University, Masaryk Hospital, Sociální Péče 3316/12A, 401 13 Usti nad Labem, Czech Republic

**Keywords:** hemodynamic monitoring, goal-directed fluid therapy, spine surgery, stroke volume, stroke volume variation

## Abstract

*Background and Objectives*: The effect of individualized hemodynamic management on the intraoperative use of fluids and other hemodynamic interventions in patients undergoing spinal surgery in the prone position is controversial. This study aimed to evaluate how the use of individualized hemodynamic management based on extended continuous non-invasive hemodynamic monitoring modifies intraoperative hemodynamic interventions compared to conventional hemodynamic monitoring with intermittent non-invasive blood pressure measurements. *Methods:* Fifty adult patients (American Society of Anesthesiologists physical status I–III) who underwent spinal procedures in the prone position and were then managed with a restrictive fluid strategy were prospectively randomized into intervention and control groups. In the intervention group, individualized hemodynamic management followed a goal-directed protocol based on continuously non-invasively measured blood pressure, heart rate, cardiac output, systemic vascular resistance, and stroke volume variation. In the control group, patients were monitored using intermittent non-invasive blood pressure monitoring, and the choice of hemodynamic intervention was left to the discretion of the attending anesthesiologist. *Results:* In the intervention group, more hypotensive episodes (3 (2–4) vs. 1 (0–2), *p* = 0.0001), higher intraoperative dose of ephedrine (0 (0–10) vs. 0 (0–0) mg, *p* = 0.0008), and more positive fluid balance (680 (510–937) vs. 270 (196–377) ml, *p* < 0.0001) were recorded. Intraoperative norepinephrine dose and postoperative outcomes did not differ between the groups. *Conclusions*: Individualized hemodynamic management based on data from extended non-invasive hemodynamic monitoring significantly modified intraoperative hemodynamic management and was associated with a higher number of hemodynamic interventions and a more positive fluid balance.

## 1. Introduction

Hemodynamic management during anesthesia is crucial to prevent fluid overload and hypovolemia. Fluid overload is potentially associated with pulmonary complications and worsened wound healing [1,2]. Hypovolemia is associated with tissue hypoperfusion and deteriorated postoperative outcomes [3,4,5].

Spine surgery in the prone position represents a particularly challenging setting from a hemodynamic point of view [6,7,8]. 

Perioperative hemodynamic instability may occur due to the prone position-induced changes in preload, afterload, and, in patients with pre-existing heart disease, decreased left ventricle ejection fraction [6,7,8,9,10]. All changes might be potentiated by the effects of anesthetic agents and blood loss. 

The impact of individualized hemodynamic management on the intraoperative use of fluids and other hemodynamic interventions in patients undergoing spinal surgery in the prone position is controversial. In a recent non-randomized study, the use of individually tailored approaches based on stroke volume variation (SVV) was shown to reduce blood loss and the use of transfusions, improve postoperative respiratory functions, and accelerate postoperative recovery in comparison to the liberal fluid strategy approach [11]. In contrast, a prospective randomized study, which used a restrictive fluid strategy in the control group and hemodynamic protocol based on pulse pressure variation values (PPV), showed more positive fluid balance in the intervention groups and no differences in blood loss or use of blood transfusions [12]. Furthermore, a before–after study on preload optimization using fluid titration according to stroke volume changes showed no difference in the amount of infused fluids [13]. None of these studies used a hemodynamic protocol based on both cardiac output (CO), stroke volume variation (SVV), and systemic vascular resistance (SVR) assessment.

Currently, continuous non-invasive extended hemodynamic monitoring is possible using the ClearSight System/EV 1000 platform (Edwards Lifesciences, Irvine, California, USA). This system continuously measures blood pressure using a special finger cuff and calculates stroke volume, SVV, and other derived hemodynamic parameters [14]. The ClearSight system measures blood pressure more reliably than traditional upper arm blood pressure cuffs [14,15]. The system has also been validated against pulmonary thermodilution, transpulmonary thermodilution, and transesophageal/thoracic echo-Doppler in cardiac output measurement; although the ClearSight system is less reliable than invasive methods, it can track hemodynamic changes [16,17,18].

This study aimed to evaluate how intraoperative hemodynamic interventions are modified using individualized hemodynamic management based on continuous non-invasive blood pressure measurements, heart rate (HR), cardiac output, systemic vascular resistance, and stroke volume compared to conventional hemodynamic management based on intermittent non-invasive blood pressure measurements. Intraoperative fluid balance was used as the primary outcome measure, whereas the intraoperative dose of norepinephrine, length of postoperative oxygen therapy, rate of wound complications, number of admissions to the intensive care unit, and length of hospital stay were considered secondary outcome measures.

## 2. Materials and Methods

### 2.1. Study Design

This single-center, prospective, parallel, two-arm, open randomized controlled pilot trial was approved by the ethical committee (approval no.: 201811 S15P) of the University Hospital Hradec Kralove, Hradec Kralove, Czech Republic (Chairperson Jiri Vortel, MD) on 23 October 2018. This trial has been registered at ClinicalTrials.gov (NCT03644654). No changes in the study protocol were made after trial commencement.

### 2.2. Selection of Participants

Adult patients scheduled for elective spinal surgery with an expected length of <3 h were considered. The inclusion criteria were as follows: age >18 years, preoperative Glasgow Coma Scale score of 15, American Society of Anesthesiologists (ASA) physical status I–III, elective spinal surgery in the prone position, and sinus rhythm on preoperative electrocardiography (ECG). The exclusion criteria were as follows: weight >120 kg, prolonged preoperative fasting >12 h, expected postoperative mechanical ventilation, preoperative hypotension (mean arterial pressure (MAP) <65 mmHg), severe respiratory comorbidity with assumed pulmonary hypertension, other than sinus rhythm on ECG, known valvular disorder, peripheral vascular disease, and assumed perioperative blood loss >1500 mL.

### 2.3. Study Protocol

Potential participants were identified on the operating list, and a screening visit was performed the day before surgery. All the patients provided written, informed consent for participation and were recruited between February and June 2019. Randomization (1:1) was performed using a computer-generated random list of patients in sealed envelopes that assigned individuals to the control or intervention group.

Standard monitoring, including non-invasive blood pressure, three-lead ECG, and pulse oximetry (SpO_2_; S/5 monitor; GE Healthcare, Helsinki, Finland), was initiated in the operating room of the Department of Neurosurgery, University Hospital, Hradec Kralove, Czech Republic. In the intervention group, a non-invasive hemodynamic monitoring probe (ClearSight system/EV 1000 platform) was used.

Anesthesia management was standardized. General anesthesia was induced with titrated intravenous propofol up to 2 mg/kg (Propofol 1% MCT/LCT Fresenius, Fresenius Kabi Deutschland GmbH, Bad Homburg, Germany) and sufentanil (Sufentanil Torrex, Chiesi Pharmaceuticals GmbH, Vienna, Austria) depending on body weight (patients with actual body weight above 60 kg received 10 µg of sufentanil; other patients received 5 µg of sufentanil) and maintained with desflurane. Tracheal intubation was facilitated using atracurium (Tracrium; GlaxoSmithKline Manufacturing S.p.A., Parma, Italy) at a standard dose (0.2–0.5 mg kg^−1^), and no additional muscle relaxants were administered during surgery. The lungs were ventilated with a tidal volume of 8–9 mL kg^−1^ predicted body weight to an end-tidal carbon dioxide (ETCO2) concentration of 35–40 mmHg. The depth of anesthesia was monitored using an entropy sensor and analgesia was monitored using the surgical plethysmographic index (SPI).

The baseline intake of intravenous fluids was standardized for both groups. PlasmaLyte (Baxter, Deerfield, IL, USA) 20 mL boluses were administered to flush the medication. Both groups received approximately 120 mL of PlasmaLyte at the beginning of anesthesia. Blood loss was compensated using PlasmaLyte with a crystalloid-to-blood volume ratio of 1.5:1.0; blood loss >1 L was compensated for through blood transfusion.

Hemodynamic interventions were targeted to maintain the MAP in the range of ±15% of the patient’s preoperative level in both groups. Hypotension was defined as a MAP <25% of the patient’s preoperative value and could be treated with the fluid bolus, norepinephrine infusion, ephedrine, or atropine (0.5 mg) in cases of bradycardia (heart rate <45 beats per minute with suspected hemodynamic impact). The fluid bolus was defined as 2 mL kg^−1^ of PlasmaLyte infused within 5 min.

### 2.4. Study Interventions

In the control group, the choice of hemodynamic intervention was left to the discretion of the attending anesthesiologist. Non-invasive blood pressure was measured at 5 min intervals, and it was recommended to shorten this interval to 2 min in cases of expected or proven blood pressure instability.

In the intervention group, hemodynamic parameters were monitored continuously and non-invasively using the ClearSight system/EV1000 platform. Data from non-invasive continuous hemodynamic monitoring were used to guide hypotension therapy and to maintain a cardiac index (CI) of at least 2.1 L min^−1^ m^−2^ (Figure 1). Atropine (0.5 mg given intravenously) was administered to patients with suspected bradycardia-induced instability.

Norepinephrine infusion (dose range: 0.02–0.1 µg kg^−1^ min^−1^) with the initial intravenous bolus of 5–10 µg was used to keep the systemic vascular resistance index (SVRI) >1900 dyn s cm^−5^ m^−2^. A fluid bolus was used in patients with a stroke volume variation (SVV) ≥9% if the CI was <2.1 l min^−1^ m^−2^ and/or hypotension occurred or persisted despite corrected SVRI. Ephedrine (10 mg intravenously) was administered to patients with hypotension or a low cardiac index and SVV <9%. Hemodynamic interventions were repeated every 5 min if the target was not reached.

Recorded variables included age; sex; body mass index (BMI); comorbidities; duration of procedure; cumulative intraoperative doses of norepinephrine, atropine, and ephedrine; intraoperative doses of propofol, sufentanil, and atracurium; the number of episodes of hypotension (an episode was defined as a maximum of 5 min of MAP <25% of the preoperative level); intraoperative fluid intake and balance; duration of postoperative oxygen administration; wound complications; length of postoperative hospital stay; and outcome. Additionally, the CI, stroke volume index (SVI), SVRI, MAP, and SVV were noted before and after anesthesia induction and after prone positioning. The complete protocol is available upon request from the authors.

### 2.5. Statistical Analysis

A power analysis based on an error of 0.05 and an error of 0.1 was performed using G*Power 3.0.9 (Franz Faul, University Kiel, Kiel, Germany). A difference of 30% in the intraoperative fluid balance with an expected fluid balance of 500 mL in the control group was considered significant for the power analysis. The sample size required for the *t*-test (two-tailed, independent samples) was calculated to be 80 patients. A sample size of 100 patients was considered to compensate for potential dropouts and possible inaccuracies in the power analysis with a planned interim analysis after the first 50 patients with predefined stopping rules (*p* <0.001 for fluid balance). All statistical analyses were performed by a blinded person.

Continuous variables are presented as the mean ± standard deviation or as the median with interquartile ranges (IQR) based on the results of a test for normality of the distribution using the Kolmogorov–Smirnov test. The *t*-test (two-tailed, independent samples) or Mann–Whitney test was used to compare results between groups based on the results of the Kolmogorov–Smirnov test. The hemodynamic variables recorded before and after anesthesia induction and after prone positioning were compared using the Friedman test with pairwise comparisons and the Conover post-hoc test. Fisher’s exact test was used to examine the differences between categorical variables. Statistical significance was set at *p* <0.05. Statistical analyses were performed using MedCalc 18.6.3 (MedCalc Software, Ostend, Belgium).

## 3. Results

A total of 54 patients undergoing elective spinal surgery were screened, and 50 patients (25 patients in each group) were enrolled in the study (Figure 2). The study was discontinued after the planned interim analysis when a predefined stopping rule was reached. The demographic and procedure/anesthesia-related data are summarized in Table 1. No significant differences were observed between the groups in any of the screened parameters.

During the procedure, the use of individualized hemodynamic management resulted in higher hypotension recognition and was associated with more interventions, namely, the total intake of fluids and the use of ephedrine (Table 2). Interventions aimed at reversing prolonged periods of hypotension (e.g., continuous infusion of norepinephrine) or bradycardia (e.g., atropine use) did not differ between the groups.

Table 3 summarizes the postoperative outcomes. There were no differences between the groups in terms of the duration of oxygen therapy, postoperative recovery room stay, hospital stay, rate of postoperative hypoxemia, wound complications, surgical revisions, or admission to the ICU. No deaths occurred during the hospital stay in any of the groups.

Table 4 presents the evolution of hemodynamic parameters after the induction of anesthesia and prone positioning in the intervention group. The prone position was associated with lower CI compared to both postinduction and baseline values. The SVI and MAP were lower in the prone position compared to the baseline values. Heart rate, SVV, and indexed value of SVR (SVRI) remained unchanged after prone positioning in comparison to both baseline and postinduction values.

## 4. Discussion

This study showed that the use of individualized hemodynamic management based on continuous non-invasive blood pressure, heart rate, cardiac output, and systemic vascular resistance monitoring significantly modified intraoperative interventions in patients with restrictive fluid management. Its use was associated with a higher number of detected episodes of hypotension and a higher number of hemodynamic interventions, including the administration of both fluids and adrenergic agonists. No differences in postoperative outcomes were observed between the control and intervention groups.

Higher use of adrenergic agents to control episodes of hypotension was recently reported in patients monitored with the ClearSight system/EV1000 platform [19]. Similar results were also recorded using other non-invasive continuous blood pressure monitoring devices [20,21]. This finding can be explained by the ability of continuous hemodynamic monitoring to detect hemodynamic changes that could be missed using intermittent monitoring methods [22]. Unfortunately, none of these studies reported better postoperative outcomes in the continuously monitored groups comprising patients undergoing noncardiac surgery [20,21,22]. The importance of improved blood pressure control, therefore, remains unclear if substantial or long-lasting hypotension is avoided [23].

The observed higher total fluid intake and more positive fluid balance in the intervention group corresponded to the results of studies on perioperative goal-directed therapy that used restrictive fluid therapy in the control arm [12,24]. In contrast, studies that used liberal fluid management in the control group generally showed a lower intraoperative fluid balance in the intervention group [11,24]. Therefore, fluid management in the control group is probably a major determinant of differences in fluid intake and fluid balance in published studies on individualized hemodynamic management. Comparison of fluid intake between studies on fluid optimization during surgery in the prone position is difficult due to differences in the patient population [11,13]. In contrast to previously published studies, our study protocol also included SVR and HR cut-off values to prevent fluid loading in patients with hypotension due to vasodilatation or bradycardia. In theory, this approach may limit the use of fluids in comparison to algorithms based solely on SV, SVV, or PPV.

The cut-off values used for SVV and CI could also have influenced the obtained results. The cut-off value for SVV of 9% using tidal volumes of 8–9 mL kg^−1^ of predicted body weight was lower than the commonly cited cut-off value of 14% [25]. This cut-off value was selected to include SVV values in the grey zone of the fluid responsiveness assessment. The used value of SVV has approximately 70% sensitivity and 70% specificity to predict at least a 10% increase in SVI after fluid loading in patients ventilated with tidal volumes ≥8 mL kg^−1^ lean body mass in the prone position [26]. The need to use higher cut-off values of SVV to predict fluid responsiveness in the prone position is explained by the observed increase in SVV in the prone position in some studies [25]. In contrast to that observation, we did not observe any significant change in SVV after prone positioning. The reasons for this discrepancy remain unclear. We speculate that the use of body support, which may have influenced chest wall elastance and venous return, could have played a role [7]. Recently, two studies using lung-protective ventilation with tidal volumes of 6–7 mL kg^−1^ of predicted body weight questioned the usability of SVV to predict fluid responsiveness in the prone position. These studies found that the use of either a tidal volume challenge [27] or a lung recruitment maneuver [28] improved the prediction of fluid responsiveness. However, the validity of these results for the tidal volumes used in this study was unclear.

The CI value used to indicate hemodynamic interventions may have also influenced our results. Although some studies [29] used a higher target of CI, typically >2.5 l min^−1^ m^−2^, our approach was in line with a recently published expert opinion [30], which suggested keeping CI >2.1 l min^−1^ m^−2^.

No differences in the postoperative outcomes were observed in the present study. In a recent meta-analysis, no mortality benefit with perioperative hemodynamic goal-directed therapy (PGDT) was observed in studies with mortality <10% [30]. A systematic review and meta-analysis, which included studies with both high- and low-risk surgery, concluded that contemporary PGDT may reduce postoperative pneumonia, acute kidney injury, wound complications, sepsis, and length of hospital stay. However, the number needed to treat (NNT) to prevent these complications ranged from 19 (wound complications) to 43 (sepsis) patients [31]. The rate of postoperative complications recorded in this study ranged from 4% to 16%. A power analysis performed for Fisher’s exact test with an alpha error of 0.01 and a beta error of 0.1, based on the rates of patients with postoperative episodes of hypoxemia (SpO_2_ <90%) or postoperative wound complications, revealed that at least 200 patients per group must be included in a possible definitive trial aimed at modifying postoperative outcomes.

Our study had several limitations. First, the hemodynamic management of patients in the control group was based on non-invasive blood pressure monitoring, which limits our ability to distinguish between the effects of continuous blood pressure monitoring and the effects of extended hemodynamic monitoring. Second, the values of hemodynamic parameters used to trigger hemodynamic interventions may be affected by the prone position, and the best cut-off values for patients in the prone position are still under debate. Other methods of fluid responsiveness assessment (e.g., tidal volume challenge or lung recruitment maneuvers) should be considered in future studies. This study was not powered to assess changes in postoperative outcomes. Finally, the single-center nature and the patient population may have limited the generalizability of our results.

## 5. Conclusions

Individualized hemodynamic management based on data from extended non-invasive hemodynamic monitoring significantly modified intraoperative hemodynamic management and was associated with a higher number of hemodynamic interventions and a more positive fluid balance. Further studies are needed to determine the role of individualized hemodynamic management in the modification of postoperative outcomes.

## Figures and Tables

**Figure 1 medicina-58-01683-f001:**
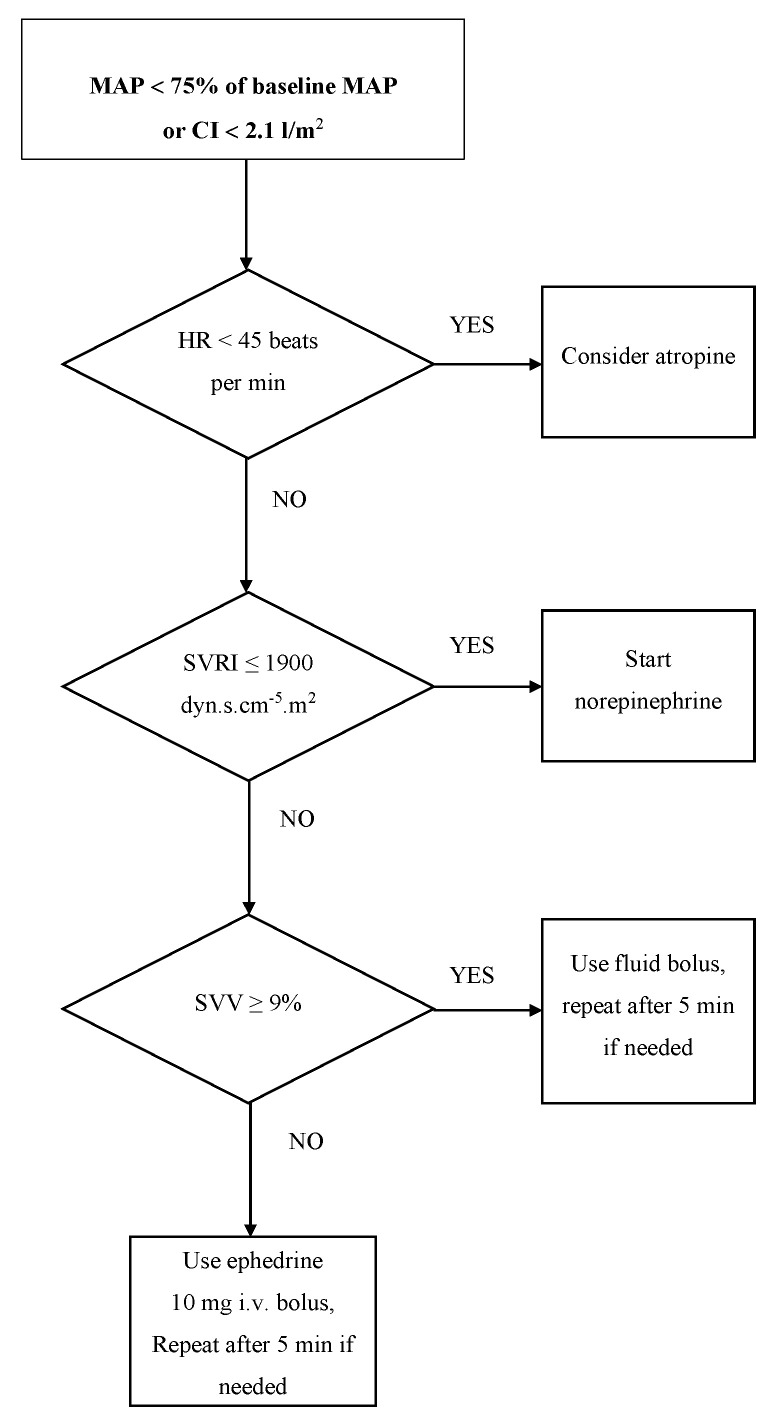
Algorithm for individualized hemodynamic interventions in the intervention group. MAP, mean arterial pressure; CI, cardiac index; SVRI; systemic vascular resistance index; HR, heart rate; SVV, stroke volume variation.

**Figure 2 medicina-58-01683-f002:**
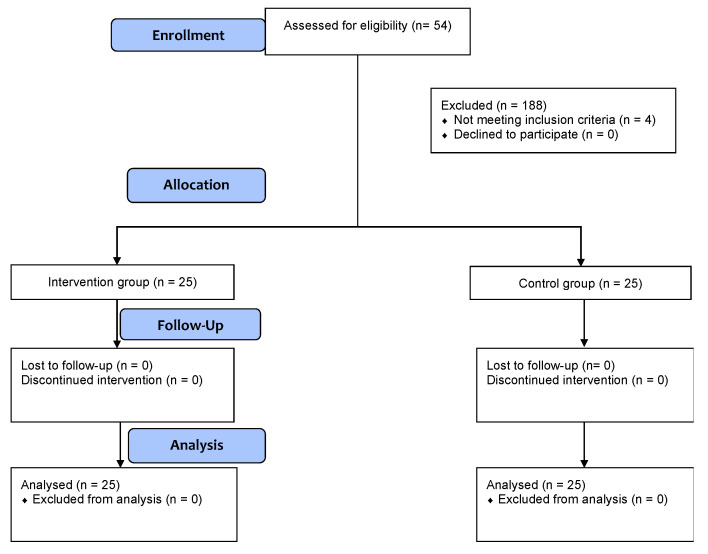
Patient allocation.

**Table 1 medicina-58-01683-t001:** Demographic characteristics, procedure, and anesthesia data.

	Intervention Group (*n* = 25)	Control Group (*n* = 25)	*p*-Value
Age, years	58 ± 15	55 ± 15	0.5297
Gender [M/F], n/n	14/11	16/9	0.7733
BMI	27.8 [25.4 to 30.2]	28.0 [24.1 to 32.5]	0.8084
ASA physical status I/II/III, n	0/18/7	0/22/3	0.1615
Type of surgery, n/% Laminectomy Decompression of a narrow spinal canal Stabilization Others	14/56% 3/12% 2/8% 6/24%	16/64% 2/8% 3/12% 4/16%	0.7733 1.0000 1.0000 0.7252
Anesthesia			
Propofol dose, mg	160 [135 to 165]	160 [148 to 200]	0.2712
Sufentanil dose, µg	30 [20 to 45]	30 [20 to 40]	0.6221
Atracurium dose, mg	20 [19 to 40]	30 [24 to 33]	0.3566
Duration of anesthesia, min	90 [75 to 115]	90 [78 to 125]	0.8648

Data are presented as mean ± SD or median [IQR]; n, number of patients; ASA, American Society of Anesthesiology; BMI, body mass index; M, male; F, female.

**Table 2 medicina-58-01683-t002:** Intraoperative hemodynamic management.

	Intervention Group (*n* = 25)	Control Group (*n* = 25)	*p*-Value
Number of detected episodes of hypotension per patient, n	3 [2 to 4]	1 [0 to 2]	0.0001
Blood loss, mL	100 [45 to 200]	100 [25 to 300]	0.7995
Total intake of fluids, mL	780 [560 to 1340]	340 [227 to 570]	0.0002
Fluid balance, mL	+680 [510 to 937]	+270 [196 to 377]	<0.0001
Number of patients receiving norepinephrine, n/%	10/40%	9/36%	0.7730
Dose of norepinephrine, µg	0 [0 to 130]	0 [0 to 275]	0.4481
Number of patients receiving ephedrine, n/%	10/40%	0/0%	0.0006
Dose of ephedrine, mg	0 [0 to 10]	0 [0 to 0]	0.0008
Number of patients receiving atropine, n/%	11/44%	12/48%	1.0000
Dose of atropine, mg	0 [0 to 0.5]	0 [0 to 0.5]	0.6936

Data are presented as median [IQR] and n = number of patients. Hypotension was defined as a mean arterial pressure <25% of the patient’s preoperative value.

**Table 3 medicina-58-01683-t003:** Postoperative outcomes.

	Intervention Group (*n* = 25)	Control Group (*n* = 25)	*p*-Value
Length of stay in the recovery room, min	120 [115 to 135]	120 [108 to 123]	0.6469
Length of postoperative SpO_2_ <95%, min	15 [0 to 49]	15 [11 to 38]	0.9840
Postoperative SpO_2_ <90%, n/%	1/4%	4/16%	0.3487
Admission to the ICU, n/%	1/4%	1/4%	1.0000
Postoperative wound complications, n/%	4/16%	1/4%	0.3487
Length of postoperative hospital stay, days	8 [7 to 10]	10 [8 to 13]	0.5619

Data are presented as median [IQR]; n, number of patients; SpO_2_, peripheral oxygen saturation; ICU, intensive care unit.

**Table 4 medicina-58-01683-t004:** Evolution of selected hemodynamic parameters in the intervention group.

Hemodynamic Parameter	Baseline	Postinduction	Prone Position
MAP, mmHg	102 [96 to 106]	97 [84 to 97]	94 [84 to 97] *
CI, L min^−1^ m^−2^	3.20 [2.50 to 3.60]	2.80 [2.35 to 3.25] *	2.65 [2.20 to 2.80] *^#^
SVI, mL m^−2^	40.5 [34.5 to 47.0]	37.0 [30.5 to 41.5] *	33.5 [29.5 to 39.5] *
SVRI, dyn s cm^−5^ m^−2^	2595 [2177 to 3319]	2850 [2229 to 3333]	2805 [2395 to 3585]
HR, beats min^−1^	77 [69 to 89]	77 [66 to 90]	74 [66 to 86]
SVV, %	11.0 [9.0 to 14.0]	10.0 [6.5 to 13.5]	9.0 [7.5 to 11.5]

Data are presented as the median [IQR]. * *p* < 0.05 vs. baseline; ^#^
*p* < 0.05 vs. postinduction; MAP, mean arterial pressure; CI, cardiac index; SVI, stroke volume index; SVRI, systemic vascular resistance index; HR, heart rate; SVV, stroke volume variation. The baseline values were recorded before the induction of anesthesia.

## Data Availability

The full trial protocol and the data supporting the findings of this study are available from the corresponding author upon request.

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
