# Peer review of "The Impact of Individualized Hemodynamic Management on Intraoperative Fluid Balance and Hemodynamic Interventions during Spine Surgery in the Prone Position: A Prospective Randomized Trial"

_medicina, 2022, doi:10.3390/medicina58111683_

Round 1

Reviewer 1 Report

I agree that it is important to introduce noninvasive monitoring method for meticulous management in surgery. From surgeon’s point of view the invasions between fixation surgery and decompression, at cervical and lumbar, are totally different. If possible, the author might concentrate on lumbar decompression surgery, which might be the largest volume. And we have to check the surgical method in the table 1. “Stabilization” is not same meaning of “fixation surgery”. Please take care of it.

Author Response

Dear reviewer,

First, we want to thank for your valuable comments. We reviewed all patients and provide more details regarding the type of surgery.

Type of surgery, N

Laminectomy

Decompresion of a narow   spinal canal

Stabilization

Others

-      Neck fixation surgery

-       

-      Extraction of metal instruments

-       

-      Revision of infected wound

-       

-      Extraction of epidural cyst

intervention 

14

3

2

               6

             (1)

             (3)

              (2)

              (0)

control 

16

4

3

4

(0)

(2)

(1)

(1)

There were minimal and of course not significant differences between types of surgery. Although we can easily add more details regarding the type of surgery in the Table 1, we suggest to keep the table (for better clarity) unchanged. Hemodynamic changes during spine surgery occur mainly due to the effects of prone position itself, effects of anaesthesia, and effects of blood loss which clearly depends on the site and type of surgery. Fortunately, the spectrum of types and sites of surgery was similar in both groups and therefore did not influence our results and drawn conclusions. We fully accept your comment to limit the spectrum of types of surgery and we shall follow this approach in our future trials.

Pavel Dostal

Reviewer 2 Report

Dear authors, 

I did find your paper interesting, but apart from clear conclusions I can see no benefit for the clinical practice.

 I order to find this paper acceptable, focus should be spread on outcomes (TOS, complications, functional outcomes etc.) so it could be relevant for clinical practice.

Author Response

Dear reviewer,

We want to thank you for your comments on our manuscript. We fully agree that we need studies dealing with clinically relevant (mainly postoperative) outcomes.

The use of extended hemodynamic monitoring could influence outcomes by either more precise hemodynamic management (and avoidance of periods associated with brain/heart/wound/other tissue hypoperfusion) or limiting excessive or suboptimal fluid intake during surgery. We were interested in the ability of the used method of non-invasive extended hemodynamic monitoring to improve the detection of episodes of hemodynamic instability (which we consider a

prerequisite for more precise hemodynamic management) and in the ability of the hemodynamic algorithm based on data from extended non-invasive hemodynamic monitoring to support intraoperative decisions about the use of fluids in patients undergoing surgery in the prone position. The results of this study support the conclusion, that the used method of extensive non-invasive hemodynamic monitoring can modify intraoperative hemodynamic management in patients undergoing spine surgery in the prone position. We believe that this information could be also considered relevant for clinical practice. We agree that the modified hemodynamic management is not automatically associated with the change of outcomes. Furthermore, outcomes depend also on the type and extent of surgery and the exact population of operated patients. Unfortunately, our study was not intended and, therefore, not sufficiently powered to detect significant differences in postoperative outcomes. We have added the acknowledge of this limitation in the Discussion and we also refer to this limitation in the Conclusions.

Pavel Dostal

Reviewer 3 Report

The article is very interesting on the actual  topic of  hemodynamic managment  during general anesthesia in the prone position.  The study  was discontinued earlier  due to the positive  balance fluid  balance in the interventional group. It is a matter of debate  whether the  cut-off values for interventions eg. SVV of 9% , MAP , Vt  etc. were not very strict. Followed by over-treatment of the patients (higher FB, ephedrine use).  The study is valueable opening new  questions on the field of perioperative hemodynamics. May be in future the design of similar study  should be apply on long lasting surgery (GA> 180 min) and follow up first 24 hours postoperatively. Note:  control on Table 2, whether dose of ephedrine, mg is 0 even in the interventional  group (0-to 10). 

Author Response

Dear reviewer,

We want to thank you for your valuable comments. When we prepared the protocol for this study, we had to decide on the cut-off values of several hemodynamic parameters that are a matter of ongoing scientific debate. We decided to follow an approach proposed by ref. 26, which was the most detailed and applicable paper on fluid response prediction in the prone position at the time of protocol preparation. We deal with this topic broadly in the Discussion (line 252 to line 268), and we believe that the used approach is still acceptable and scientifically correct, though newly published data suggest using other methods to improve the prediction of fluid responsiveness in the prone position.

Continuous hemodynamic monitoring enables the detection of short-term blood pressure fluctuations that could be unrecognised in patients monitored using intermittent blood pressure measurement. Therefore, we believe that the observed differences are not indicators of overtreatment but markers of more precise hemodynamic management. We also followed your suggestion to check the results in table 2. The results are correct – in the intervention group, ephedrine was used only in 10 out of 25 patients, and the median has a value of 0.

We agree with your comment that future studies should use either more gentle indicators of postoperative outcomes, select a more severe population, or use a substantially higher number of patients, and we shall follow these recommendations in our future studies.

Pavel Dostal